# Pandemic-Induced Qualitative Changes in the Process of University Studies from the Perspective of University Authorities

Vida Navickiene [1] , Valentina Dagiene [2],*, Egle Jasute [2],*, Rita Butkiene [3] and Daina Gudoniene [3]

1   Faculty of Creative Industries, Vilnius Gediminas Technical University, Sauletekio Al. 11, LT-10223 Vilnius, Lithuania; vida.navickiene@vilniustech.lt
2   Institute of Educational Sciences, Vilnius University, Universiteto Street 9, LT-01513 Vilnius, Lithuania
3   Informatics Faculty, Kaunas University of Technology, Studentu Street 50, LT-51392 Kaunas, Lithuania; rita.butkiene@ktu.lt (R.B.); daina.gudoniene@ktu.lt (D.G.)
*   Correspondence: valentina.dagiene@mif.vu.lt (V.D.); egle.jasute@fsf.vu.lt (E.J.)

**Abstract:** The pandemic COVID-19 period in education has brought many challenges to all organizations. The activities of the higher educational institutions are being affected and the situation may last for a long time. Under the current circumstances, it is important to shift to distance learning through online processes and improve educational processes at all organizational levels. Institutions have to ensure successful distance or remote learning process by identifying their opportunities, meeting challenges, and establishing the sustainable quality factors for remote or distance learning. This study aimed at identifying the pandemic-induced qualitative changes in studies that have occurred at the levels of university authorities, lecturers, and students. Universities of Lithuania were taken as a case study. The novelty of the research lies in the fact that the focus of analysis is not on the negative effects of the pandemic observed in higher education studies but on finding positive qualitative changes that are also of importance to future studies. Phenomenographic qualitative research strategy was chosen in the research and 15 in-depth semi-structured interviews with experts in university studies were conducted. Seven categories were distinguished during the research representing qualitative changes in studies at three levels—authorities, lecturers, and students. The discussed levels seemed to have a mutual effect on each other. The external motivation of leaders and the support and establishment of work and online study conditions encouraged both external and internal qualitative changes in studies from the perspective of lecturers as well as students.

**Keywords:** higher education; qualitative changes; distance learning; university authorities; lecturers; students

## 1. Introduction and Background

Due to the pandemic situation, educational institutions have met many challenges regarding their organization and implementation of distance learning. This has become particularly apparent during the COVID-19 pandemic when higher education institutions were forced to transform their studies and to make them available online. In this situation, the universities have played an important role in supporting stay-at-home society and have been a valuable addition to their productive home environments. Under such rapid changes, the universities should be able to monitor the quality of their studies [1]. The success of studies depends on the decisions of university authorities and lecturers and on their possessed experience and motivation to ensure high quality of transformations in studies. Due to a sudden pandemic and a shift to online learning, many lecturers have not had adequate time to adjust to the new teaching platforms [2,3]. "Educators, learners and institutions have had to adjust to these changes and manage the various increased demands related to workload, new practices and external regulatory authorities" [4]. The

pandemic has stimulated changes and transformations; therefore "a new and transforming attitude is being created, which will change the fundamental essence of universities" [5]. Education is changing its transmission methods, and e-learning will undoubtedly become a vital strategy moving forward [6].

Due to the pandemic, many educational institutions have started to search for effective and alternative ways of learning by using online platforms that universities provided, such as Moodle, Microsoft Teams, Zoom, or others; however, many challenges were encountered, especially in delivery of STEM subjects. Many researchers analyzed various kinds of tools to facilitate learning environments when using online systems, for example, Tencent Meeting System [7,8] or Topic Analysis Instant Feedback System [9]. The number of online course offerings has been increasing significantly, but recent trends have shown a steady progression in the normalization of online studies. There have been periods when higher education was fully transformed and moved online. However, it is not clear whether this transformation produces positive study outcomes.

Over the last years, numerous research studies embracing different areas have been conducted where researchers have analyzed challenges evoked by the pandemic, emerging difficulties or experienced negative effects. Some of them have particularly emphasized and criticized instances where distance learning has prevented students from obtaining practical skills and adapting to the labor market [10–13]. Others have noted that some academics themselves lack knowledge of information technology and online teaching [4,7,14,15] and that teaching and learning resources adapted to pure online education are scarce [1,16,17]. Some others have emphasized problems related to poor mental health, which emerges due to social exclusion and uncertainty of the future situation [18–20]. Adjusting studies to online learning has posed various challenges to educational institutions. The main question that is of interest to many educators and education policymakers is whether online learning is better and more effective than class-based learning [10,21–24]. It can be stated that contact and distance learning will always have their supporters and opponents. However, one or another kind of learning is only a form filled with various tools. Thus, the quality of managing and using such tools depends on willingness to act and the decisions of participants in tertiary education: university authorities, lecturers, and students. On the basis of research results the article attempts to show positive qualitative changes, most of which were created through appropriate decisions of university authorities. The research results can be used in other higher education institutions as good examples of how to improve the quality of the study process.

The novelty of our article is related to its analysis of what qualitative changes were preconditioned by the COVID-19 pandemic. Thus, the article presents the results of research, which disclose the positive consequences of the pandemic. The emerging challenge was accepted with difficulty. Initially, there was a lot of uncertainty, fear, and unawareness of how to act. At that stage, the leadership of university authorities, timely decision-making, and close collaboration with the academic community were of utmost significance. Higher education as an institution is distinguished by slow change compared to other organizations. This is partially due to subordination in decision-making: faculty administration, rectorate, senate, and the university council. However, the most considerable changes during the pandemic have been preconditioned by the determination of the academic community and their internal desire to act. Despite the pandemic situation, which has created many issues for higher education institutions, some positive effects and newly raised opportunities have been recognized. Several innovative approaches and tools for learning online have been developed. Study resources have been revised, restructured, and adapted for students' self-directed learning. Academics and students have been fostered to improve their educational and digital competences. These changes and opportunities have created a space for innovative thinking and innovative solutions [5,12,25–28]. The e-learning quality is the most significant aspect of students' e-learning, constituted by the quality of e-learning tutors, the course material, and the e-learning administration and support services [29]. Although this situation is in line with the vision and mission

of future learning in the era of industrial revolution 4.0 and community 5.0, it still has advantages and disadvantages. However, the current freedom cannot be interpreted as unlimited freedom in learning [30]. Universities must develop innovative ways to deliver teaching without compromising on quality [13].

The pandemic of COVID-19 has forced higher education institutions to rapidly adapt to new conditions and ensure the transition of the study process into distance education. This type of transformation has led to qualitative changes in the study process and opened new opportunities gradually emerging from a stressful and unstable situation. The majority of researchers support the idea of online studies due to a number of reasons, which can be referred to as qualitative factors, e.g., the e-learning systems offer many advantages and compensate for the weaknesses of the traditional learning methods. For example, there are new approaches and tools for capacity development [21,31]; online learning can reduce costs without reducing the quality of learning [32]; and high-quality participation can improve the breadth and depth of student's learning [33]. Therefore, higher education institutions have to adapt their study programs to respond to the needs of the transitional period [34].

Although education has globally transformed and moved online due to the pandemic, it is unknown whether this transformation produces more positive teaching and learning outcomes or negative issues. Therefore, this research problem is formulated as the following question: what qualitative changes in studies were preconditioned by the pandemic from the perspective of the university?

## 2. Research Design and Methodology

All the Lithuanian universities terminated studies due to the first wave of COVID-19 pandemic on 16 March 2020. The majority of universities had a two-week break, which was used for preparation to implement distance studies. The biggest university was the only one to continue studies without any preparation. Such a different path to distance studies had a certain different impact on the quality of studies (more of negative character), but the research shows that later these differences disappeared. Thus, the research also included the universities that chose a different strategy of transition from contact to distance studies.

Lithuania, a small country with a population of over 2.5 million, joined the Bologna Process in 1999 and became a member of the EU in 2004. Therefore, its national system of higher education is in line with other countries of the Bologna Process. The higher education system in Lithuania is of binary character and consists of two types of higher education establishments, i.e., universities (study duration—4 years) and colleges (study duration—4 years). The first university was established in Lithuania in 1579 and the foundation of the second one was laid much later, in 1922. The majority of Lithuanian universities started their activities in the middle of the 20th century and, therefore, higher education in Lithuania is rather young. Aiming to increase accessibility of higher education, colleges were established in 2000. However, in contrast to some European countries, the degrees of master and doctor can be obtained only in universities. Thus, these institutions offer broader education not only in terms of content but also in forms. Furthermore, colleges in Lithuania are more focused on implementing regional educational policy, whereas universities respond to national needs. Colleges are established mainly in regions and universities are concentrated in cities.

It should also be mentioned that in 2019 the requirements for external assessment of study programs changed and since then considerable attention has been allocated to systemic improvement of teacher competences and student support. This is also reflected in the research results because experts speak a lot about improvement of teacher competences and centers established in institutions that provide support to them.

There are 11 state universities in Lithuania. They can be divided into three types (as provided for in the national legislation)—firstly, classical universities (2), which implement studies in all the study areas and are the oldest universities in Lithuania; secondly, universities of broad profile (5), which carry out studies in several areas; and thirdly—specialized

universities (4), which target the arts, music, sports, and military sciences [35]. Specialized universities are small in terms of numbers of students and academic staff members as well as the structure of their faculties. Therefore, the volumes of data they can provide for the research are not big. Some study fields offered by them are also available in universities of broad profile.

The research design. The research was carried out from 15 to 30 June 2020. This time period was chosen for the research as it allowed investigation of the challenges posed to the study process by the pandemic, problems encountered by the participants in the study process, and decisions of authorities made in the unstable situation, as well as identification of qualitative changes predetermined by the pandemic.

The research question: what qualitative changes in studies were preconditioned by the pandemic from the perspective of the university? A phenomenographic strategy was chosen to answer the research question. The methodology of phenomenographic research calls for diverse experience of research participants. Therefore, informants with various experiences were included in the research. Conducting the research, criterion sampling was used for selecting informants. The main criteria considered were as follows: (a) informant is from one of the biggest Lithuanian universities (size); (b) from different types of university: classical university and broad profile universities from different towns (type); (c) leaders are available in the university, whose functions are related to the management of studies (taking highest positions in the university, then leaders of lower rank (deans of faculties of humanities, social, technological, engineering sciences, heads of other divisions, who were chosen to make sure that the position of authorities is the same in terms of the range of qualitative changes influenced by the pandemic); (d) informants—experts in their field with at least 7 years of managerial experience related to the analyzed phenomenon (experts, working experience). Therefore, informants in the article are referred to as experts (see Table 1). The triangulation of experts aimed to ensure the validity of the research.

**Table 1.** The encoding of experts.

| Experts | Codes Are Seen Describing the Research Results | Gender |
|---|---|---|
| 3 vice-rectors of studies of different universities | 1INF–1VR, 8INF–8VR, 12INF–12 VR | 3 men |
| 3 study directors of different universities | 2INF–2SD, 11INF–11SD, 13INF–13SD | 1 men, 2 women |
| 4 deans of social faculties | 5INF–5DS, 6INF–6DS, 9INF–9DS, 14INF–14DS | 2 men, 2 women |
| 3 deans of technology faculties | 4INF–4DT, 7INF–7DT, 15INF–15DT | 3 men |
| 2 directors of different departments | 3INF–3DD, 10INF–10DD | 2 men |

All the experts were master's degree holders, and the majority had a doctor's degree.

The process of studies depended on decisions made by authorities; they managed the whole process of university studies and received information from all the lower divisions. Moreover, vice-rectors saw the situation all over Lithuania because they belong to the Conference of Rectors of Lithuanian Universities, whose members made strategical decisions at the national level during the pandemic. Thus, the experience of such experts is very important for solving the research problem. They were chosen as critical cases, whose attitudes, actions, and activities predetermined further qualitative changes in the study process. Qualitative research was conducted, which included 15 semi-structured in-depth interviews. The verbal consent to take part in the interview was received from all the experts. Conducting the last interviews, the research became saturated with information because experts started repeating information. Thus, it was pointless to take interviews with more experts.

The interviews with experts were done using the Zoom video conferencing platform. The interviews lasted from 45 min to 75 min. The total duration of all the interviews was about 940 min. The received data were decoded and about 130 pages of decoded material was received. To ensure the validity of interviews, they were carried out by two researchers and the received data were processed by three researchers. MAXQDA was used for interview transcription analysis.

The phenomenographic research not only identifies the categories that describe the phenomenon under consideration but also reveals hidden, tacit meanings and presents their interrelationships. The distinguished categories form a hierarchical horizontal order. Data analysis identified descriptive categories and the outcome space of the concept expressed by a network of logically related, hierarchically organized, and systematized categories [36,37]. Authors [38] present the space of results of the investigated concept as a logically structured set of different ways of experiencing a phenomenon. The space of results expresses experiences and discloses internal relations among described categories. The links of horizontal levels that are of the same level are applied in the research [39]. The analysis of research results allowed distinguishing seven categories, which characterize study-related qualitative changes resulted in by the pandemic (see Table 2). The research participants were authorities of universities, who pointed out changes they noticed or even influenced in universities. Conducting the analysis of interviews and applying the methodology of Kinnunen et al. [40] revealed changes at three levels: authorities, lecturers, and students. The table was devised following the methodological approach of Kinnunen et al. [40].

The research data were obtained in line with research ethics and summarized information on respondent affiliation to the city, the university she/he represented, or the current position held.

**Table 2.** The categories characterizing study-related qualitative changes resulted in the pandemic at the levels of university authorities, lecturers, and students (according to Kinnunen et al., 2007 [23]).

| Categories | How Are the Qualitative Changes Understood in the Study Process? | What Is the Research Focus? | Dominating Aspect |
|---|---|---|---|
| Changes in forms of authorities' work | Possibility of working and organizing meetings of university authorities online. | What essential change in the study organization did representative authorities experience? | Authorities |
| Establishment and maintenance of mutual/parity relation-based relationship between the authority and the university community | Close communication of authorities with the university community (heads of lower-level divisions, lecturers, students) and strong consideration of community opinion. | How did authorities encourage the community to actively engage in the qualitative process of study transformation? | Authorities |
| Provision of academic and technical support to the community | Timely and continuous academic support (preparation of new training courses) for lecturers and their technical provision. | What main actions of authorities influenced study-related qualitative changes? | Authorities |
| Supply of new forms of studies | The emergence of a wider variety of study forms: blended master's studies, hybrid learning form for national and international students, virtual/blended mobility. | How did the synergy of collaboration between the authorities and the community influence the structure of studies? | Authorities |

**Table 2.** *Cont.*

| Categories | How Are the Qualitative Changes Understood in the Study Process? | What Is the Research Focus? | Dominating Aspect |
|---|---|---|---|
| Mastering new tools of distance studies | The new experience acquired by lecturers encouraged innovative studies, preparation of high-quality material for distance studies, the possibility of improving the quality of the content of study subjects, application of elements of distance learning in traditional studies. | How did support provided by authorities encourage qualitative changes in studies? | Lecturers |
| The internal potential of lecturers | Empowered/encouraged intrinsic motivation and self-confidence of lecturers organizing the study process was the basis for qualitative changes. | What influence did external support of authorities and motivation have on lecturers as the main organizers of studies? | Lecturers |
| Internal turnover of students | Transformations in studies and distance learning led to improved attendance of students and they improved skills of independent learning. | What essential qualitative change occurred at the level of students? | Students |

## 3. Research Results

Our research is focused on three key factors identified in education and raised in the pandemic period, i.e., (1) qualitative changes at the level of authorities, (2) qualitative changes at the level of lecturers, and (3) qualitative changes at the level of students.

### 3.1. Qualitative Changes in Studies at the Level of Authorities

**Changes in the form of authorities' work.** During the pandemic, the global shift to distance learning also affected the work of university leadership. Due to the pandemic, they were forced to work in other forms, and to chair and hold online meetings of university authorities. *"One of the good experiences of this is that we clearly understood that organizing administrative meetings remotely has numerous advantages. The ministry has finally realized how much time it saves on its meetings [...] after the quarantine ended, this practice remained and now almost half of the meetings, if not more, are organized remotely. The quality of solutions has not really diminished"* (8VR). Meetings and consultations remotely using a video conferencing platform have led to closer communication between the central university management and faculty administration. *"I really appreciate communication in the community because we seem to continue to have more frequent meetings, for example with vice-deans"* (1VR). Such change in the form of work enabled authorities to see that distance work is possible. Since the pandemic delimited everybody and divided the university into many local places, the leaders had to find ways and to assume leadership mobilizing the community for common work.

**Establishment and maintenance of mutual/parity relation-based relationship between the authority and the university community.** Mastering video conferences encouraged communication and collaboration with all of the community. The university leaders did it very actively and at different levels. Quality changes preconditioned by the pandemic period of COVID-19 include the opportunity to cooperate in decision-making and to maintain communication between management and the community. This is emphasized by all the experts and they all list different links, means of communication, or goals.

First, the communication occurred between the different levels of leadership: central administration, faculty deans, vice-deans, and heads of departments: *"Of course, teamwork*

*was important [...] Discussing a variety of things, sharing good practices, what works well, what could be done differently*" (2SD). "*I very positively evaluate this communication of community because we are likely to continue having such meetings more often (e.g., with vice-deans.*" (12VR).

Secondly, it can be assumed that communication between the university leadership and lecturers was promoted because of the necessity of managing the situation and explain to them "*why particular decisions have been made*" (1VR). This necessity stimulated positive changes in collaboration. "*There have been a lot of efforts to keep the communication with the lecturers on all issues. They are the people that influence the study process and its quality the most. There has been a great deal of effort to inform them about all sorts of different decisions. Perhaps it was not always done directly, but rather informing through the heads of departments or deans*" (3DD). Communication takes place to not only explain decisions and exchange best practices, but also to ensure a positive psychological climate and maintain team spirit and established traditions. It was also important to stay in constant contact with students, who experienced challenges of distance learning during the pandemic. While communicating with students, it is important to explain not only the decisions that have already been made but also to communicate clearly, explain things that are not clear, and let them know when to expect further information. They were told how "*distance lectures work and how they should behave during them*" (1VR) as well as "*principles of academic integrity*" (13SD). In addition, various recommendations have been drawn up for distance learning. During the pandemic the gaps in communication with students were identified, communication became more consistent and targeted, and new communication channels were opened, which should remain and be maintained in future as well.

It should be emphasized that the biggest change, the transition to working remotely, and the other factors of change listed above were a consequence inspired by the collaboration of the academic community. The experts acknowledge that this is also one of the drivers of change, as everyone felt focused and saw the close focus of the academic community in trying to manage the situation. "*We learned that our lecturers are actually advanced. They started sharing additional tools and instruments and used them creatively. There were certainly those who then voluntarily sent their own materials to other faculties*" (14DS). 2SD: "*The situation showed that the academic community is focused and that even in uncertain and extreme conditions, the university is able to function, to be able to ensure the study process*" (2SD). Thus, challenges encountered during the pandemic not only united the community for the common goal to smoothly move to distance learning, to ensure quality studies, but also encourage collaboration outside the faculty and entering the space of the whole university.

Third, quality studies and their success during the quarantine were determined by the fact that the central or faculty management responded to opinions of lecturers conducting distance studies, rapidly addressing the problems and carrying out prevention. "*After each lecture, lecturers used to write to us, the Directorate of Studies, the vice-rector or the dean, or the Academic Support Centre about the difficulties, what kind of training they wanted or what problems were, or even shared good practices if the lecture succeeded*" (8VR). This provision of teaching experience to the central management ensured the preparation of temporary documents regulating the study process, which covered a wide range of study programs offered by different faculties. It can be stated that the synergy of collaboration emerged due to the pandemic, which introduced positive changes to the further process of studies. The documents regulating studies were not only revised again but also grounded on lecturers' practical experience. Thus, confidence between the leadership and lecturers was created or enhanced.

**Provision of academic and technical support to the community.** The research shows that the leaders provided academic and technical support, which affected the most important quantitative changes—improvement of lecturers' competences. First, the results of the study show that more than half of the experts emphasized the creation of an additional online website/section on the website that publishes all necessary, constantly updated information. "*Methodological and technical information for lecturers, advice for students, the information provided to scientists, administrators and all members of the community. Topical issues*

*are systematically presented, updated so that people can have a single point of access*" (11SD). These qualitative changes in study organization are of long-term character and the community will be able to use them in the future as well.

During the pandemic, two types of training courses were organized: on tools of distance learning (*distance work training for lecturers)* and didactics (*provision of didactic)*, or how to use these instruments in the study process. Most experts stressed the importance of providing lecturers with initial and then continuous support by teaching them how to master distance learning tools. According to the experts, "*The E-learning Technology Centre conducted training weekly, sometimes even twice a week, on how to use tools, also prepared instructions for using Zoom, Teams*" (13SD); "*There have been specific training on how to prepare tests, how to make some recommendations regarding final theses, etc.*" (7DT).

In addition to training and e-mail information, lecturers were given various recommendations to be used when preparing or conducting distance lectures and these recommendations were broad in scope. "*Afterwards those instructions and short videos were very helpful*" (4DT).

Thus, qualitative changes induced by the pandemic had a long-term effect on lecturers' didactic and ITC competences, positively influenced the work of students, and enhanced the lecturers' self-confidence.

Maintenance, creation, or improvement of the technical base also brought a positive qualitative change in studies resulted in by the pandemic. The view could be taken that the university authorities had to reallocate the available resources and to allocate finances to the assurance of high-quality distance studies. Improving/developing a technical base is one of the most important quality factors. First, it was decided to purchase Zoom licenses for videoconferencing enabling lectures longer than 40 min: "*We understood the need and had to buy additional licenses abruptly to make better use of the lecture*" (12VR). Secondly, other necessary technical means were provided, and the need was learned from the faculties: "*Information used to be passed on and the faculty was told to identify the needs, how much of the hardware employees had including cameras, microphones, etc.*" (15DT). Third, some universities made decisions regarding additional rooms to record lectures for the purposes of the faculty: "*We bought and prepared two recording studies: one on the X Street, where we have a building and the other one right here, so it is easier for lecturers to record lectures*" (6DS).

**Supply of new forms of studies**. The pandemic-induced qualitative changes in the studies are also related to the changing structure of studies. The representatives of all universities stated that students have the opportunity to study in a different form of master's degree in the form of blended learning: "*That is why we have made a decision regarding master's studies from the first day of September next year (2021). Totally, 50% of the activities in the program can take place remotely. Two days of learning remotely, two days of regular contact*" (1VR). It should be noted that one university has opened up even wider possibilities by offering distance master's studies in the regions: "*Here we are talking about a regional master's degree, where it has already been planned to take place in a distance way. It was planned to take place in the regions directly, then rescheduled to remote*" (5DS). This form of blended learning provides a greater opportunity for postgraduate study and ensures better attendance at classes. It could be argued that postgraduate studies do not have a lot of laboratory work, but are research-oriented, so the necessary experiments for the final work can be performed on a more flexible schedule.

Secondly, other forms of study and access to them are open to foreign students. For many lecturers, especially in fields of technology and engineering, it is an innovation to use a hybrid form of learning. Prior to the pandemic and the quarantine, hybrid learning was mostly applied when a higher education institution has faculties in different cities. Quarantine has shown the possibility that, with the necessary equipment, it is possible to organize such studies even for students abroad. Foreign students choose this option because they cannot come to Lithuania due to the quarantine: "*We will give lectures to foreigners online, first-year full-time lectures and exercises in foreign English will be given to those students who have arrived in person, and the lecture or exercises will be broadcast remotely in*

*real-time. The lecture is attended online by those students who will not have the opportunity to come. Similar measures will be taken with the postgraduate studies.*" (12VR). People in isolation could also take advantage of this opportunity.

Another opportunity for expanding internationalization is virtual /mixed mobility. It is an opportunity for students to acquire/improve their intercultural competences in other ways. "< . . . > *these concepts will also appear in our new documentation. The European Commission has confirmed that such virtual mobility exists officially. This is the case when students do not go abroad but still study at a foreign university*" (13SD). Virtual mobility creates the possibilities and conditions for halting the decrease of international exchanges and reduces the threat of weakening intercultural competence.

### 3.2. Qualitative Changes in Studies at the Level of Lecturers

**Mastering new tools of distance studies.** The research shows that for many lecturers, working remotely was a completely new activity because of a widespread belief that the quality of the study or subject was suffering from it. The pandemic forced everyone to take up teaching activities in a distant way and opened up new opportunities, leading to innovative studies: "*In these two weeks, we have learned more than ever before*" (15DT). The experts say this is one of the things that helped to speed up a lot of other processes: "*Lecturers appreciated that distance learning and distance learning tools are really useful, they can be used with a purpose in mind. It is a great help to lecturers to organize the learning process in more ways and to allow students to learn in more diverse ways. Of course, distance learning should be just one form of learning that complements contact form and diversifies it*" (8VR).

The study has found that the development of high-quality distance learning materials has made a significant contribution to the successful transition from contact learning to distance learning and has resulted in long-time qualitative changes in studies: "*Records, additional tests, additional materials, various external sources improved the study quality*" (15DT). The Moodle environment used by Lithuanian universities has been greatly supplemented by subject material prepared by lecturers and adapted for distance learning: "*At this point, the amount of lecture material in Moodle has been strongly supplemented*" (4DT).

The use of virtual environments has revealed the possibility of improving the quality of learning content: "The study content will improve, since learning material will be uploaded to the virtual learning environment every year and you will be able to update and improve your course every year" (6DS). This has also been noticed by university management that is considering how to motivate lecturers to do so: "It is possible to motivate by offering additional vacation days, financial motivation, it can be encouraged in other ways, but distance courses need to be prepared." (9DS).

Mastery of new tools of distance studies in the future will contribute to more significant qualitative changes in the study process. Most experts emphasize that lecturers who have tried and already have distance learning experience will increasingly want to apply elements of distance learning in traditional studies: "*It is now clear that activities in a virtual learning environment will become more frequent*" (6DS); "*Lecturers plan to move at least part of their subjects to a distance form.*" (13SD). In order to ensure the quality, the lecturers will start preparing for the next semester in advance. It should be emphasized that the lecturers see an opportunity not only to organize the studies more by applying the elements of distance learning but also to use them for organizational work.

**Internal potential of lecturers.** Although the studies during the quarantine period posed many challenges, they also opened up new opportunities and even induced internal qualitative changes. They can be attributed to studies because lecturers' internal potential is directed to student teaching. One of the most important factors is the intrinsic motivation of lecturers. Various opportunities open up when confidence or readiness to deliver distance teaching is gained. This is primarily due to the fact that distance learning was tested and used by all lecturers working during the semester, and not individually as before the quarantine. The internal concept and desire of the lecturer promote the external study processes of the higher school. Thus, massive intrinsic motivation has been gained.

Lecturers have gained confidence in their own strengths and in presenting distance learning opportunities for their subjects; trying new ways of teaching and communicating with students; and mastering tools. Such internal changes related to lecturers through mastered tools of distance studies will encourage qualitative changes in the process of studies: "*We will be using a lot more and bolder elements that we have tested and that work well. Lecturers will no doubt feel much more confident in those processes after testing both the study methods in a variety of ways and the assessment methods they saw paying off. Finally, they will definitely feel much more confident in organizing the process itself.*" (2SD). Thus, it can be stated that compulsory distance learning will reduce the gap between contact and distance learning, as it has encouraged the use of certain distance learning tools and even changes in certain levels or forms of study.

The obtained intrinsic motivation encouraged lecturers to exchange their good experiences. This was carried out either on a university-wide basis or within the faculty. Several experts emphasized their importance and positive impact on the studies. "*They organised such workshops, shared experiences or carried out remote training*" (8VR), "*We did internally, at the initiative of our own lecturers who are advanced in the use of technology*" (14DS). It should be noted that if training to work remotely usually took place at the beginning of quarantine on general issues, then the exchange of best practices was usually organized later when lecturers had already acquired experience and self-confidence. A common issue for all lecturers was how to organize student examinations; therefore, "*Many people were very active in sharing ideas with each other on how to organise examination and testing, whether to take tests or use other types of written tasks*" (15DT).

Empowered intrinsic motivation of lecturers provided opportunities both to lecturers and to students (even more to the latter).

*3.3. Qualitative Changes in Studies at the Level of Students*

Although there were a lot of discussions regarding the process of managing the situation by transitioning to distance education in the event of a pandemic, thus ensuring the continuity of the quality learning process, such types of learning during the quarantine have led to some changes. Many elements of traditional studies have also changed as a result of the emergency situation and the transition from face-to-face to distance learning. Qualitative changes at the student level are best seen through their inner changes.

**Internal changes in students**. The majority of experts point out that student attendance has improved. Several experts mentioned this positive change, although some of them said that it was true only for a while and others claimed that it was a permanent change. It is entirely possible that this was due to the specifics of the subject and the lecturer's ability to involve students in the active study process. However, almost everyone referred to good attendance: "*Increased attendance is considered to be a success. Attendance increased up to 100%, which was not the case during regular lectures*" (14DS). Only one expert saw a temporary improvement: "*After the study process was restored, the increased attendance lasted for about two weeks. In the third week, attendance fell sharply. Since distance learning was a novelty, everyone wanted to see how things were here. Thinking in the long run that distance learning would lead to better attendance, the answer is no because it was just an effect of innovation*" (12VR).

With regard to profound change, emphasis should also be placed on developing students' self-directed learning. Several experts noticed that the unexpected and even forced transition to distance learning encouraged students to learn more independently and develop learning-to-learn competences: "*Lecturers saw that students are forced to actually study in this way and they have a lot of work to do on their own*" (15DT), "*Students began to realize the importance of independent work. Until then, apparently, no one even thought, say, even in terms of hours, how much of that work should actually be. There was an opinion that everything should be learnt during the lecture*" (14DS).

## 4. Discussion

The transformation in tertiary education posed not only challenges but also promoted qualitative changes. New challenges associated with online teaching and learning will create a space for innovative thinking and innovative solutions [5]. The COVID-19 pandemic forced university communities to acquire determination and test distance learning and work. However, the most considerable qualitative changes in the uncertain situation were predetermined by the decisions of university authorities and the new forms of work tested during the pandemic. Decisions arrived at during the pandemic have an impact on the future and, for this reason, technical solutions also have to be considered [15]. Four main quality factors, which require attention, can be distinguished: economic, psychological, social, and environmental ones [32]. Changing the model of management "requires working with existing organisational cultures to ensure the collaborative participation of educators and learners throughout the process" [41]. Communication is one of the most significant factors that preconditions qualitative changes in times of uncertainty [41,42]. It is important to notice that the successful change was due to a focused and collaborative academic community across the university, and not just at the faculty level. Like in the case of face-to-face learning, conscious online learning communities should be established as well [16,43]. Belonging to a community is of utmost importance in distance learning, as is the development of meaningful relationships with one's instructors and classmates and having goals and interests similar to groupmates [44]. Students should be aware of reasons for such changes and how they can accept this [14].

The transfer of the study process to online learning meant that ways of maintaining relations between the lecturers and students, teaching and learning methods, etc. had to undergo changes as well. University leaders were forced to reconsider their actions to make them efficient longer than the transition (pandemic) period and to ensure the quality of this change in the future. The main role here is played by lecturers, who have to create a friendly environment for students through technology-based teaching [17,32,45,46] and maintain the quality of interaction between the lecturer and student [47]. This was one of the difficulties, which forced all the related lecturers to learn themselves. The attention was directed to online pedagogy [1,25]. Namely, here the university authorities made suitable solutions and provided academic and technical support to the community. A flexible and supportive online learning environment was able to fight against social isolation and increase social participation, but lecturers needed constant help to enable them to do this practically [16,23,43,45,48]. Lecturers' ability to work and teach online is one of the success factors and qualitative changes. For this reason, it is necessary to continue and even expand such training courses [34,49]. Students' assessment posed the most serious challenge both to lecturers and students [45]. Various training courses offered on time were very valuable and had residual value because they were attended by a big number of lecturers. Lecturers will need to engage in novel methods to achieve effective teaching outcomes, which may affect the quality of tertiary education [3]. Lecturers' ability to use the newest technologies in online teaching can be a significant factor, which can either encourage or hinder student and professor usage of e-learning. For these reasons, professors are expected to be more facilitators, collaborators, mentors, trainers, directors, and study partners and provide choices and greater accountability for students to learn [32].

However, changes have also brought some positivity as well. It should be mentioned that new opportunities that emerged have been initiated by the increased intrinsic motivation of lecturers. They have gained self-confidence and prepared (learned) to teach remotely. Therefore, deep reasons (intrinsic motivation of lecturers) primarily influenced the implementation and development of distance studies not only during the pandemic period but also after it. Lecturers will want to apply the elements of distance learning in traditional studies, when writing or defending their final project. They have also opened up opportunities and have shown that the work they put in during the pandemic is valuable and sustainable, as the quality of subject content can be improved annually.

Striving for qualitative changes creates a need to strengthen the IT platform by making necessary changes with respect to its continuous availability and uninterrupted services [14]. Users' personal factors have no direct influence on user satisfaction, while platform availability has the greatest influence on user satisfaction [7].

The transformation in tertiary education has encouraged the appearance of new study forms at universities or strengthening of already existing ones. "The move to on-line learning may stimulate an increase in blended and more accessible forms of education and teaching styles have had to change and this may have a lasting effect" [25]. The newly opened opportunity to organize various new courses only partially depends on the encouragement and support of university leadership. Lecturers' intrinsic motivation and acquired competences serve as the most relevant contribution. "Online education and its success lie in the participants and their qualities" [46].

COVID-19 has forced us to attempt to enhance student experiences and learning outcomes via online rather than proximate learning [5]. Students were dissatisfied with many things: accessibility, social, lecturer issues [50], and time costs, physical, and mental work in front of a computer screen [32]. That is why it is necessary to create a support group for members working in different areas to manage the situation in the university [17] and to provide constant support to students [49]. Distance education must be intelligently combined with face-to-face teaching because the student relationship with the professor is essential, and distance education, paradoxically, strengthens it [32]. Faculty and teaching assistants need to provide students with timely feedback, including online video tutoring and email guidance after class. It is necessary to adopt some measures to improve the degree and depth of students' class participation [33].

The main factors influencing user satisfaction with the online teaching platforms were system quality, interaction quality, service quality, and platform availability [7]. The move to distance learning brought innovation and better quality of study [5]. For lecturers, this has involved the experience of mastering the latest methodology and tools of distance learning [51]. Due to online teaching and learning, both students and teaching staff will further develop their online communication and interpersonal skills through regular exposure to online platforms [25]. All the lecturers in the future will have an opportunity to have intensive meetings with students and to co-create learning outcomes via online platforms [5]. The acquired self-confidence in ITC competences enhanced lecturers' intrinsic motivation to further use distance learning platforms. Such learning is acceptable for many students as well and their attendance increased even to 100% [45]. They consider distance learning a good idea and have plans to use it more often during the semester [23]. The e-learning and online students node include one relevant theme, i.e., self-regulation of students [52]. Namely, online learning encouraged students' self-discipline and self-education [51]. "Aspects of student characteristics, intrinsic motivation, teacher/lecturer characteristics, infrastructure, system quality, course quality and information, and an online learning environment guarantee existing learning success" [53]. For students to maximally benefit from online learning contexts, online courses must be designed to support students' self-regulation because students no longer have reinforcements commonly found in traditional face-to-face learning contexts [28]. By studying autonomously, students can easily apply a learning approach that aims to self-regulate both their own motivation and desires and the expectations they want to achieve. Such learning leads to the personal life of everyone in viewing learning for himself as having responsibility, as a control for the acquisition of their knowledge [24]. Motivation and self-regulation also played a role in successful online learning. Online students were more predisposed to self-study, self-discipline, and regulate their time management. [54]. Students actively create unique learning experiences by using their own environment and resources [26]. COVID-19 confinement changed students' learning strategies to a more continuous habit, improving their efficiency as well as enhanced their inner responsibility [55].

## 5. Conclusions

The conducted qualitative research highlighted three levels (authorities, lecturers, students), where pandemic-induced qualitative changes occurred in the university. The first level of leaders had the most significant influence on qualitative changes related to lecturers and students. In the preparatory phase, shifting from face-to-face learning to online one and later, they were the first to test and become convinced of the reliability of new forms of work. The possibility of working and organizing online meetings regarding studies became one of the qualitative changes not only for the future but also provided conditions for the establishment and maintenance of mutual/parity relation-based relationship between the authorities and the whole university community. It should be acknowledged that communication with heads of departments at different levels, lecturers, and students lacked systematicity in the pre-pandemic period. It was based more on traditional events. During the pandemic, the university authorities assumed the role of leaders to mobilize the community and to encourage successful transfer and continuation of high-quality studies online. Communication was one of the reasons for success. The community members appreciated consideration of their opinion preparing temporary documents, constant relation was ensured, and decisions were explained. The establishment of communication and close relation is the second qualitative change, which can have an influence on further collaboration and nurturance of communication culture. The actions of university authorities providing timely and continuous support to the academic community should be emphasized. Their encouragement and support resulted in systemic learning of new innovative online tools among lecturers so that they are able to maintain a relationship with students. It can be stated that this is a double qualitative change—due to decisions of authorities, training courses of online didactics and ICT were introduced in the university, which enhanced the learning culture of university lecturers and universities became learning organizations. This is important to lecturers as well because they were forced to learn and master new innovative tools of online studies. This knowledge and abilities will be used in the future as well. Moreover, this also contributes to the fourth significant qualitative change in studies—appearance and implementation of new study forms for learners. The development of study forms ensured better accessibility of higher education to different groups of the society.

The new experience acquired by lecturers mastered tools of online teaching and learning promoted innovative studies, preparation of high-quality study material, the possibility for improving the quality of subject content, and application of elements of online learning in the traditional face-to-face studies. This external qualitative change in studies, which was encouraged and supported by the university authorities, empowered and enhanced the intrinsic motivation of lecturers and their self-confidence and opened up new opportunities for innovative studies. Thus, a paradoxical phenomenon can be observed, when pandemic-induced changes forced all the lecturers to learn to pursue the qualitative transformation of studies. However, this obligatory activity did not evoke much resistance from the lecturers. On the contrary, this external act had an influence on intrinsic motivation and encouraged qualitative changes in studies. Intrinsic motivation is the strongest driving power of innovation and change.

The most essential change at the level of students was that they not only embraced the shift to online learning and gained the ability to do this but also improved their attendance and strengthened their ability to learn independently. It can be assumed that having perceived the difficulty of the situation they worked hard to cope with all the challenges. This qualitative study-related change is important to lecturers as well. Reflecting on their experience, they should continue using online tools and other methods that promote students' independence.

Thus, all the discussed levels were not separate aspects of qualitative change in university education, but rather they influenced each other. The external motivation of university leaders, their support, and the creation of conditions for online studies led to external as well as internal qualitative changes in studies for lecturers and students.

The obtained research results have significant empirical importance for planning and maintaining the study process of high quality. It has to be emphasized that the obtained research results are of value for the study process not only during the time of the pandemic. A part of the described processes, such as establishment and maintenance of mutual/parity relation-based relationships between the authority and the university community, as well as timely and continuous academic and technical support to the community, are of paramount importance for a quality study process under varying work conditions and forms. It can be stated that the heightened prevalence of distance learning and work during the pandemic not only will retain its current value but also will expand with a growing supply and availability of such studies. Therefore, not only universities in Lithuania but also communities of other universities can make use of ways and forms of supporting the academic community. For example, motivating and empowering lecturers to work online and to implement the presented content to students in a qualitative way.

There are some limitations of this study. On the basis of attitudes of experts-representatives of university authorities, who participated in the research, the qualitative changes were formulated. Lecturers and students, whose opinions and experience could supplement the acquired research results, did not participate in the research. Only considering the experience of the latter, the experts formulated their attitude. Such research would be one of the priorities of future research. Moreover, only the qualitative changes observed in distance studies during the pandemic are presented in the article. Although the experts were asked about the encountered problems and difficulties learning or working online, these issues were not formulated as part of the goal of this article. Future tasks for researchers could include comparing qualitative changes and encountered problems or challenges as well as suggesting solutions. During the pandemic the whole academic community was forced to work and learn online. Therefore, one more limitation should be pointed out—to what extent the research conclusions could be similar or different under traditional study conditions, when face-to-face learning prevails, and to what extent the distinguished qualitative changes are maintained after returning to the usual form of work when the pandemic is over. Such a research question could be answered by conducting continuous research in the future.

**Author Contributions:** Conceptualization, V.N. and V.D.; methodology, V.N. and R.B.; formal analysis, R.B., E.J. and D.G.; writing—original draft preparation, V.N., V.D. and D.G.; writing—review and editing, R.B. and D.G.; visualization, V.D. and E.J.; project administration, D.G.; funding acquisition, D.G. All authors have read and agreed to the published version of the manuscript.

**Funding:** The research is supported by Lithuanian Research Council financed project "Model of distance working and learning organization and recommendations for extreme and transition period" (EKSTRE) (01-06-2020-31-12-2020). Grant Agreement S-COV-20-20.

**Institutional Review Board Statement:** Ethical review and approval were waived for this study, as this study involves no more than minimal risk to subjects.

**Informed Consent Statement:** Informed consent was obtained from all subjects involved in the study.

**Data Availability Statement:** The data presented in this study are available on request from the corresponding author. The data are not publicly available due to data restriction policy by the grant provider.

**Conflicts of Interest:** The authors declare no conflict of interest.

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
