# Peer review of "Pandemic-Induced Qualitative Changes in the Process of University Studies from the Perspective of University Authorities"

_sustainability, doi:10.3390/su13179887_

Round 1

Reviewer 1 Report

Although interesting and captivating, your paper contains a number of issues / suggestions for improvement that I would like to raise:

  • Although partially formulated, the purpose of your research should be CLEARLY and UNAMBIGUOUSLY stated in the introductory section. Additionally, a theoretical gap can perhaps be better unveiled.
  • Why is worthy of focusing exclusively on the positive outcomes of the pandemic? That mentioned, your research to some extent acts as an advertisement for the online learning.
  • As I comprehend it, your research addresses only remote or online forms of learning in the dichotomy between “e-learning” and “face-to-face learning”. That accentuated, can a hybrid learning be a promising solution?
  • What makes the Lithuanian higher education sector an interesting case?
  • A succinct description / presentation of the Lithuanian tertiary education sector is deemed pivotal.
  • Did you use an interview guide while interviewing your informants? If so, it would be nice to place it in an attachment. What kinds of survey data (line 170) did you use?
  • Why did you emphasize (target) “the levels of university authorities, lecturers and students”? Are some other levels missing in your research? Did you conduct interviews with lecturers and students? If so, I am very much willing to hear their voices as well. Are views expressed by administrators (e.g. vice-rectors) not politically biased? Perhaps more importantly, can they objectively reflect those of lecturers and students?
  • Are 15 interviews sufficient enough to portray the situation at all 11 universities in Lithuania? Have all the universities faced the similar challenges? In other words, are your findings generalizable to all 11 higher educational establishments in Lithuania?
  • Please make it crystal-clear what your main (theoretical and/or empirical) contribution to the current body of knowledge is.
  • What are the limitations (deficiencies) of your research and proposals for further studies?

Reviewer 2 Report

The article is well worked in all the phases of its design, preparation and realization. It would be convenient to establish a table with a description of the profile of the participants, as well as a graph of the relationship of the coinciding arguments of the three groups of participants, so that it is more clarifying to the reader.All references should be reviewed and adapted to the journal format.
